# "The Ark of Rights": Development of a Board Game to Empower Older Adults Regarding Their Rights

Carla Sílvia Fernandes [1,*], Camila Neto [2], Catarina Silva [2], Sara Dionísio [2], Susana Oliveira [2], Isabel Amorim [3], Alice Delerue Matos [4,†] and Maria Manuela Martins [1,†]

[1] Research Center for Health Technologies and Services (CINTESIS), Porto Higher School of Nursing, Rua Dr. António Bernardino de Almeida, 4200-072 Porto, Portugal
[2] PSI-ON, Association for Education, Development and Intervention in Communities, Avª João XXI, No. 1027, Vermoim, 4770-768 V.N. Famalicão, Portugal
[3] European Anti-Poverty Network/Braga Centre, 4700-251 Braga, Portugal
[4] Department of Sociology, and Communication and Society Research Centre, Institute of Social Sciences, University of Minho, Campus of Gualtar, 4710-057 Braga, Portugal
* Correspondence: carlafernandes@esenf.pt
† These authors contributed equally to this work.

**Abstract:** There is an urgent need to ensure the rights of older adults. In particular, there is a lack of awareness of human rights by older adults themselves, for which intervention strategies should be developed. Due to the need for intervention at this level, a board game was created to empower older adults regarding their rights using a dynamic and interactive method. This article aims to describe the development stages of the board game "The Ark of Rights"® up to its pilot study. Its development followed three stages: A first phase to review the scientific literature and benchmarks on the rights of older persons, a second phase to define the game design and collect statements from older people for the game, and a third phase to test the game. The European Portuguese Validation of the System Usability Scale (SUS) was used to assess the latter phase. Approximately 200 older people contributed to the game's contents (second phase), and 74 participated and positively evaluated the game's usability and their satisfaction with its use (third phase). In summary, the game "The Ark of Rights" revealed itself to be a resource for empowering older adults regarding their rights. It also enables the identification of possible human rights violations among older adults and subsequent intervention.

**Keywords:** aging; human rights; games; elder abuse





## 1. Introduction

In today's world, and particularly in industrialized countries, we are faced with increasing average life expectancy resulting from the evolution of medical science and the progressive generalized improvement of living conditions. As a result, by 2050, the UN estimates that there will be 1.55 billion people worldwide aged 65 and over, representing 15.8 percent of the population [1]. Thus, the global proportion of people aged 65 and over will approximately double in the next thirty years, making older people the fastest-growing demographic group in the world [1,2].

Aging is a natural phenomenon that inevitably occurs in the human life cycle and brings challenges to older people's lives, mostly resulting from changes in their bodies, mind, thought process, and lifestyle [2,3]. This process gives rise to stereotypes and discrimination against this group, which is often seen as a single, undifferentiated, and homogeneous group [4].

Due to their increasing frailty, older adults are a vulnerable group and may be disadvantaged when needing to exert pressure to claim their rights [5]. In theory, everyone should have full and equal enjoyment of human rights, but this is often not the reality for older people [6].

In recent years, progress has been achieved in the incorporation of norms and programs focusing on older adults [7,8]. However, it is noted in the various victim support structures that the numbers known are not even close to the real number of older adults who are victims of crime [5,9].

There is a lack of awareness of human rights among older adults and those who care for them [10]. However, it is fundamental to involve older adults and place them as active subjects with the ability to demand the fulfillment of their rights [5]. Given that one of the difficulties in guaranteeing human rights in this population is the general lack of awareness of human rights per se [10], and being aware of the need to intervene at this level, we took the initiative to develop a game with the purpose of empowering older adults about their rights.

The use of games has been spreading and is increasingly asserting itself as an important strategy to awaken new intervention possibilities [11–13]. With the use of games, a predominantly paternalistic approach is abandoned, and there is a paradigm shift toward facilitating the person's self-efficacy instead [12]. The design and development of a game imply a concern for multiple items. Dynamics in a gamified resource include the use of constraints, emotions, narratives, progression, and relationships [11,14]. Mechanics consists of a set of rules that determine the outcomes, and dynamics are the users' response to the set of these mechanics [11]. Creativity and repeated play with the rules lead to outcomes between the game and its players that determine the game's dynamics; these dynamics can be designed to achieve certain goals [11,14]. Simple participation in a game can promote a change of behavior and/or knowledge of the user [11–13].

Games must be carefully planned in order to achieve the desired results, involving rules and competition, and often require props or other devices [12,14]. Game development involves various stages. The first phase is crucial to conceptualize the contents to be included in the game, then the game design is defined, and finally, the game is tested in the field [15].

The objective of the game described below is to dynamically and interactively empower older people regarding their rights. It is also an intervention tool that enables the mediator to identify possible rights violations and uncover situations of abuse and violence against older people. Considering all the above, it seemed relevant to contribute to the knowledge in this area. Therefore, in the present article, our objective will be to describe the development stages of the board game "The Ark of Rights"®.

## 2. Materials and Methods

In this paper, we describe a pilot study that aims to describe the development stages of the board game "The Ark of Rights"®.

### 2.1. Study Procedure

In the first phase, a review of the scientific literature and benchmarks on the rights of older persons was carried out. The aim was to identify the state of the art on the rights of older persons and the existing material on educational games to address human rights with older people. No publications on educational games related to this topic were found, justifying the development of the following stages. In the second phase, the game design was defined. In this phase, the game mechanics and rules were chosen, determining the way participants interact [15]. The game cards were designed based on the information collected from nearly 200 older people who reported situations in which their rights were respected or, on the contrary, disrespected. Their contents and categorization were based on the UN Principles for Older Persons [16].

This phase implies the use of gamified strategies with easy configuration, constituting the design of the prototype [11]. In this phase, it should be defined whether cooperation or competition will be used; these aspects of game mechanics dictate how players interact [15,17]. The mechanics determine the game's purpose and how the participants achieve it. The choice

of mechanics is crucial for gameplay and should make sense to the players. In addition, aesthetics are also important for game mechanics [14].

In the third phase, the game was tested. The European Portuguese Validation of the System Usability Scale (SUS) was used. In total, 74 older persons participated and evaluated the game's usability and their satisfaction with its use.

### 2.2. Participants

The game "The Ark of Rights" was developed in the context of a group called Grupo de Capacitação Interconcelho na área do Envelhecimento (Intermunicipal Training Group in the area of Aging), coordinated by EAPN Portugal/Centro Distrital de Braga (District Center of Braga). The entities promoting the game are EAPN Portugal/Centro Distrital de Braga and PSI-ON–Associação para a Educação, Desenvolvimento e Intervenção nas Comunidades (Association for Education, Development, and Intervention in Communities). The following partner entities also collaborated in the development of the game: Associação de Paralisia Cerebral de Braga (APCB), Cruz Vermelha Portuguesa–Delegação de Fafe, Escola Superior de Enfermagem do Porto, Fraterna–Centro Comunitário de Solidariedade e Integração Social and Instituto de Ciências Sociais da Universidade do Minho. The game was funded by the Velh@ Amig@ project, co-financed by POISE/CIG under typology 3.16 Financial and Technical Support to Non-Profit Civil Society Organizations, which falls under the responsibility of PSI-ON.

For the construction phase of the game cards, the 8 experts included in the writing of this article took part, integrating the testimonies of nearly 200 older persons and adopting different disciplinary perspectives.

For the evaluation phase of the game, a convenience sample of participants from daycare centers, social centers, and the community was used; 74 questionnaires were collected. Data collection for the construction phase of the game and its evaluation took place between July and December 2022. The inclusion criteria were the older adults aged 65 years or older, conscious and oriented in time and space (application of the MMSE-Mini Mental State Examination Test for older people aged 65 years or older), who agreed to voluntarily participate in the study after signing the informed consent. Older adults with cognitive deficits (e.g., Alzheimer's disease, Dementia), severe visual impairment (unable to be corrected using glasses/lenses), severe hearing impairment (unable to be corrected using a hearing aid), and with a clinical indication for not performing the activity were excluded.

### 2.3. Instruments and Data Analyses

The data collection instrument consisted of a first part on the brief characterization of the participants, followed by the application of the European Portuguese Validation of the System Usability Scale (SUS), and a last part consisting of an open question with free-text responses. At the same time, the technicians were asked to observe the application of the game.

Regarding the resource of the game usability instrument, SUS, although it was developed for technological resources, it has been used in other strategies, namely in the development and use of physical games. SUS consists of statements that evaluate the degree of usability of the game through ten claims, ranging from 0 to 50, with a midpoint of 30 [18]. The statements in the scale allow its applicability to resources other than just technological ones.

A descriptive statistical analysis was used to analyze the closed questions, and content analysis, according to Bardin, was used for the open questions [19].

### 2.4. Ethical Considerations

The research was in line with ethical precepts and was approved by the Research Ethics Committee in May 2022 (ADHOC11132022). Throughout the study, the participants' anonymity was maintained, and the confidentiality of the data obtained was ensured. In data analysis, data coding was used to ensure confidentiality. The data coding is known

only by the researchers. In addition, security measures were implemented to protect the information.

## 3. Results

### 3.1. Creating the Content of Cards

The cards were built based on the United Nations Principles for Older Persons [16]. In the first phase, 180 cards were prepared and divided into four types: Positive Narratives (+): 60 cards, Negative Narratives (−): 60 cards, Challenge: 30 cards, and Rights: 30 cards.

Interactive dynamics squares, considered special squares, were included on the board. For this purpose, four interactive dynamics were identified for insertion into the board: Positive Communication (3); Mobility (3); Affective Interaction (3); and Emotional Expression (3).

### 3.2. Pre-Test of the Game

A pre-test was carried out in different social centers in the country's northern region with around 100 older people. This phase served to fill in existing gaps and suggest possible changes, namely shorter stories, only alluding to one human right on each card, and the use of more simplified language. In relation to the game board, based on the results of this step, the number of squares in the game was reduced because its application would be too long. Thus, the pre-test phase determined the reduction in the number of squares (from 75 to 22): special card squares (from 12 to 4), positive narratives card squares (from 11 to 5), negative narratives card squares (from 16 to 5), rights card squares (from 20 to 3), and challenges card squares (from 10 to 3). Along with these changes, a revision of the contents was carried out. The revision focused on the contents and the language used in the cards.

### 3.3. "The Ark of Rights" Boardgame

The game mechanics allow six to eight players to participate. The players start by rolling the dice to determine their starting order. The player with the highest score starts the game, followed by the player to his or her right, and so on. Each player uses a pawn of a specific color. The players throw the dice and move their pawns on the board according to the number shown on the dice. The board game consists of four colored squares (blue, pink, purple, and orange).

Each color corresponds to a pack of cards: Blue cards—positive narratives (+); pink cards—negative narratives (−); purple cards—challenges; and orange cards—rights (Figure 1).

The narrative cards describe situations in which the rights of older people have been respected (blue cards) or, on the contrary, disrespected (pink cards). They aim to promote debate on the situation described, and the mediator is responsible for stimulating this debate. The challenge cards are designed to promote interaction between players by carrying out certain dynamics. The mediator can also use these cards as an ice-breaker. Finally, the rights cards cover issues concerning the rights of older people according to the United Nations Principles for Older Persons: Independence, Participation, Care, Self-Fulfillment, and Dignity [16].

On the game board, there are four special squares, duly marked, which refer to the following activities: positive communication; mobility; affective interaction; and emotional expression.

The "Positive Communication" square is based on the principles of laughter therapy [20]. Games that trigger emotional responses can be powerful behavior and learning tools [15]. In the "Mobility" square, the facilitator chooses one or two groups of exercises according to the ability of the group. It is desirable that the mediator exemplifies and invites everyone to do the movements, repeating each one five times.

In the "affective interaction" square, the player should give a hug or a warm and expressive greeting to another player. The mediator can invite everyone to participate by hugging or greeting another player [21].

In the "Emotional Expression" square, the player must compliment another player. The mediator can invite everyone to participate by complimenting another player [22].

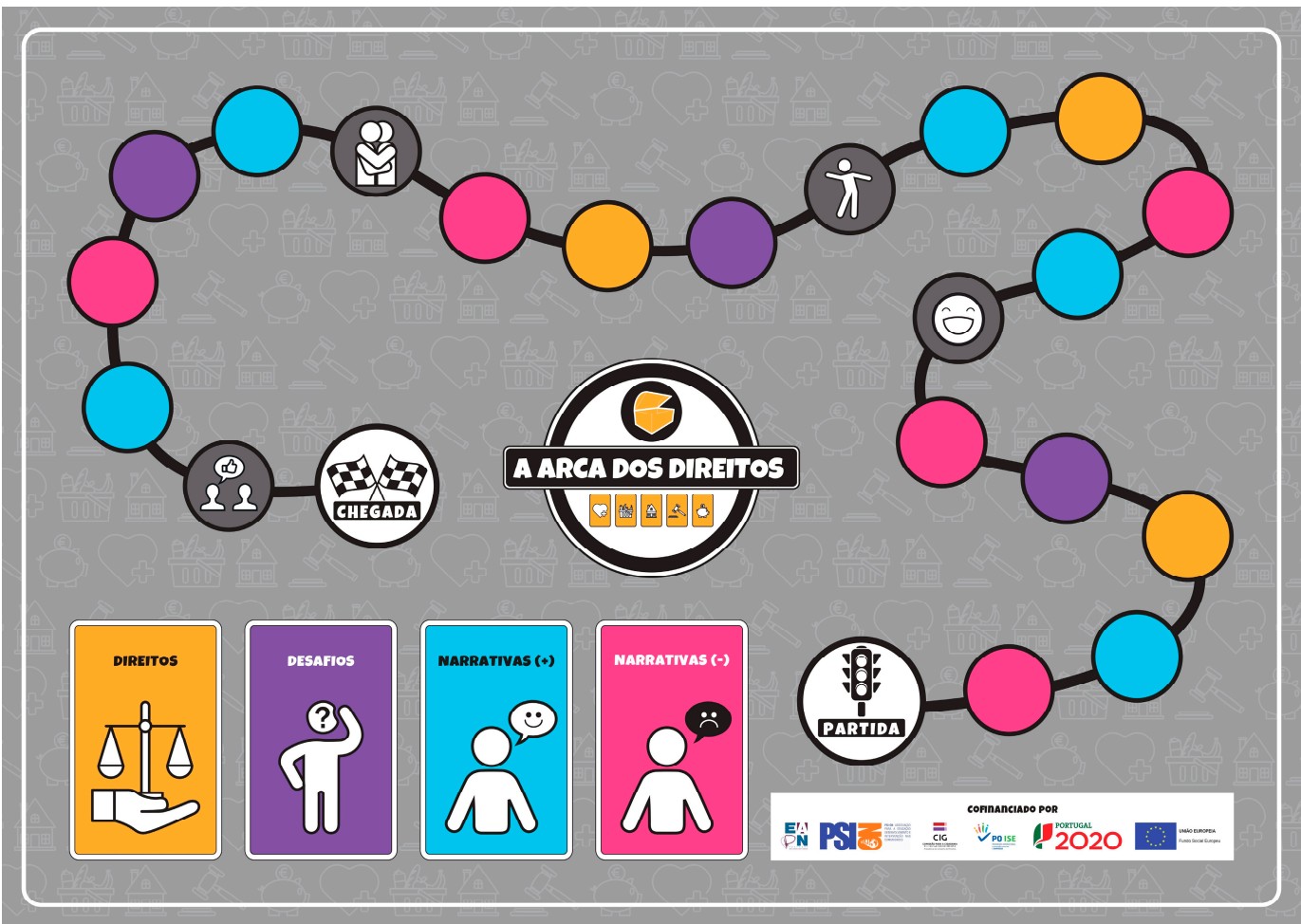

**Figure 1.** "The Ark of Rights" board game.

The game ends when one player reaches the final square on the board. The mediator should explain what the intervention consists of and its objectives. Each player must roll the dice and control their pawn's movement on the board. The mediator should only help players if they have difficulty performing these tasks. The cards used in the game should be read by the mediator, who, if necessary, can adapt the language so that everyone understands them.

### 3.4. Application and Evaluation of the Game

There were 74 accepted responses from the game participants; 32.4% were male, and 67.6% were female. The age of the participants ranged from 65 to 93 years, and the mean age was 74.07, with a standard deviation of 10.391. The education level was distributed between 'can neither read nor write' and, in equal percentages, 'Basic Education-3rd cycle' (4.1%), 'Can read and/or write' (24.3%), 'Secondary Education' (1.4%), and the most frequent situation was 'Basic Education-1st cycle' (58.1%).

In terms of marital status, 45.9% were widowed, 33.8% were married, 9.5% were single, 5.4% were married, and 5.4% were in a consensual union. The places of data collection were: day centers (37.8%), homes (28.4%), senior academies (20.3), and other places (13.5%).

Figure 2 shows the degree of agreement with the items of the SUS scale. The overall scale score ranged between 27 and 50, with an average of 38.7, which is higher than the mean value of the scale, thus validating its degree of usability.

From Table 1 it can be seen that the best averages were for item 1 "I think I would like to use this game often," for item 9, "I felt very confident using this game," and item 3, "I found the game easy to use."

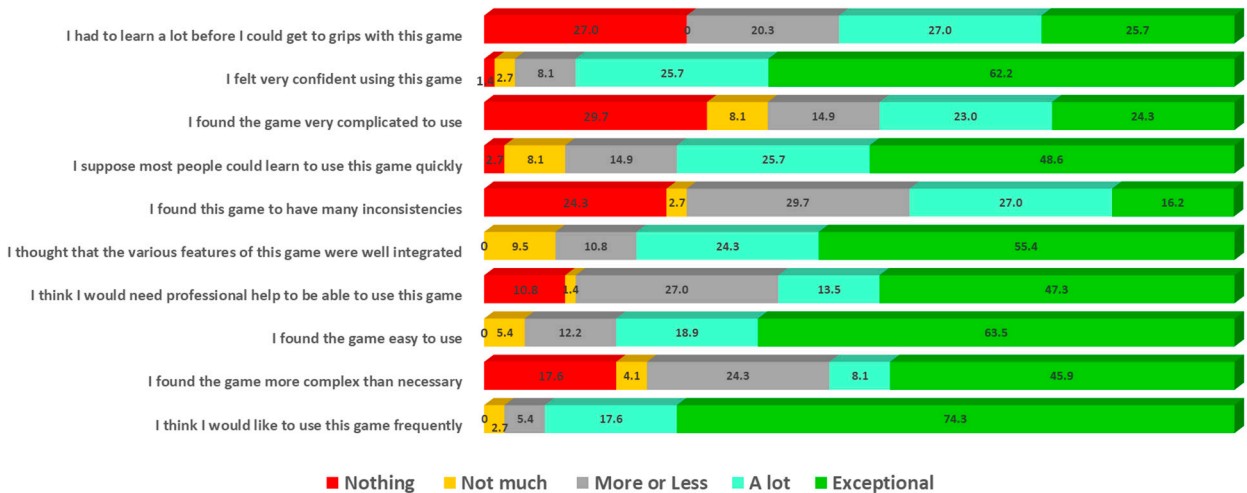

**Figure 2.** SUS items score %.

**Table 1.** Statistic parameters SUS analysis.

| Items | Min | Max | Mean | SD | Kurtosis | Asymmetry | |
| --- | --- | --- | --- | --- | --- | --- | --- |
| | | | | | | Statistic | Standard Error |
| 1 | 2 | 5 | 4.64 | 0.713 | −1.544 | −2.123 | 0.279 |
| 2 | 1 | 5 | 3.61 | 1.524 | −1.080 | −0.616 | 0.279 |
| 3 | 2 | 5 | 4.41 | 0.905 | −1.260 | −1.360 | 0.279 |
| 4 | 1 | 5 | 3.85 | 1.331 | −1.057 | −0.902 | 0.279 |
| 5 | 2 | 5 | 4.26 | 0.994 | −0.229 | −1.144 | 0.279 |
| 6 | 1 | 5 | 3.08 | 1.392 | 0.315 | −0.336 | 0.279 |
| 7 | 1 | 5 | 4.09 | 1.100 | 0.135 | −1.080 | 0.279 |
| 8 | 1 | 5 | 3.04 | 1.583 | 0.752 | −0.154 | 0.279 |
| 9 | 1 | 5 | 4.45 | 0.862 | 3.433 | −1.804 | 0.279 |
| 10 | 1 | 5 | 3.24 | 1.533 | 4.231 | −0.446 | 0.279 |

From the analysis of the free-form responses, the following stood out: I liked it very much (*n* = 12), I understood very well what the game consisted of (*n* = 7), I would like to play more often (*n* = 6), it was a moment of conviviality and distraction, but with utility (*n* = 4), it was very quick, and there were still many cards (*n* = 2), and no comments (*n* = 43).

## 4. Discussion

The game "The Ark of Rights"® was developed within a group called Grupo de Capacitação Interconcelho na área do Envelhecimento (Intermunicipal Training Group in the area of Aging), coordinated by EAPN Portugal/District Center of Braga. The entities promoting the game are EAPN Portugal/District Center of Braga and PSI-ON–Associação para a Educação, Desenvolvimento e Intervenção nas Comunidades. In addition, the following partner entities also collaborated in the development of the game: Associação de Paralisia Cerebral de Braga (APCB), Cruz Vermelha Portuguesa–Delegação de Fafe, Escola Superior de Enfermagem do Porto, Fraterna–Centro Comunitário de Solidariedade e Integração Social and Instituto de Ciências Sociais da Universidade do Minho. The game was funded by the Velh@ Amig@ project, co-financed by POISE/CIG under typology 3.16 Financial and Technical Support to Non-Profit Civil Society Organizations, which falls under the responsibility of PSI-ON. The development of this game required a structured design consisting of several stages with well-defined objectives to achieve the proposed results.

Games must be planned carefully to enable the expected results to be achieved [14]. Thorough design thinking should include considerations of game mechanics, dynamics, aesthetics, emotions, and contexts of the game and players [15], which was achieved with the board format of *The Ark of Rights*. Games can be used as a successful intervention tool for effective behavior change [12,15]. However, a number of aspects need to be considered, such as how participants interact within the game—with each other and themselves [15]. It is also important to determine whether the game's interface and systems are intuitive and easy to use [14].

Games need to be tested to expose possible errors and conduct any necessary reformulations [12]. The ability of games to achieve behavior change is a product of their design—to address a clearly defined problem [15]. This stage of playtesting is all about getting people to come and play the game to see if it engenders the experience for which it was designed [14]. The satisfaction and usability of the game were enhanced by the use of the special houses in this game, adding an emotional component to the game. The application of the SUS scale, although not designed for this purpose, showed an average of 38.7, which is higher than the average value of the scale, thus validating its degree of usability.

The System Usability Scale (SUS) is a widely used survey tool that measures the perceived usability of a system or product. It is a reliable and valid method for assessing users' satisfaction with a system's usability [18] and can be applied to a board game. By measuring these aspects of usability, the SUS can provide valuable feedback to improve this game's usability and user experience. This step is useful for the later validation phase of the game impact.

In a time very much linked to technological resources, physical games, such as card or board games, may be considered unelaborate or retrograde and often devalued. However, as physical games are cheaper and easier to produce, they remain a viable alternative for knowledge and/or behavior change interventions. At the same time, they constitute an inclusive resource and promote social participation through the gaming experience [15].

Physical games, such as card and board games, can be a suitable resource in this group by improving interpersonal communication, interpersonal relationships, self-efficacy, self-care, and decreasing loneliness [22], as well as enhancing the determined goals behind the game. An important aspect of any game should be a focus on creating engaging experiences and increasing interactivity between participants [11,17]. Using a board game can facilitate interpersonal communication and relationships by relying on group learning, which leads to exchange, interaction, and connection among participants [12,22].

Furthermore, this approach supports their responsibility to exercise their participation in the political process and other aspects of their community life [7,9,16]. This focus is clearly visible in the objects that guided the implementation of the "The Ark of Rights" game.

The entities who integrated the construction of this resource have an interest and need to ensure the rights of older people with whom they deal on a daily basis, namely, to jointly find answers to improve the provision of care to older people, review their professional and institutional practices, with a view to continuous improvement in order to promote the physical, psychological and social well-being of older people [15].

This study has some limitations, namely the convenience sampling strategy. Therefore, the results should be interpreted with care when considering the transferability of the results. In addition, the medium and long-term impact of the game on the target population has not yet been evaluated.

Despite these limitations, we consider that this study has implications for fostering reflection on this topic. Our results also allow us to ensure usability and satisfaction with the use of the game, observable in the scores of the usability scale and the comments of the participants.

## 5. Conclusions

In summary, the board game "The Ark of Rights" proves to be a promising resource to empower older people about their rights in a dynamic and interactive way. It also allows it to be an intervention tool for identifying possible violations of rights and subsequent intervention in situations of abuse and violence against older people. Older persons are entitled to enjoy all human rights fully and invoke the general guarantees in human rights treaties. The results show that the game helped construct a new strategy, providing the opportunity for players to reflect on their rights. Thus, it can serve as an important assessment and intervention strategy.

Throughout this article, we aimed to describe and highlight the steps necessary for developing a game. We know that studies with interventions on this thematic area are still very scarce, but there is value in continuing to assess and develop interventions that enhance the human rights of older adults.

## 6. Patents

Co-authored work registration No 2953/2022-THE ARK OF RIGHTS®.

**Author Contributions:** Conceptualization, C.N., C.S., S.D., S.O., A.D.M. and I.A.; methodology, C.S.F., C.N., C.S., S.D., S.O., A.D.M., M.M.M. and I.A.; formal analysis, A.D.M. and M.M.M.; investigation, C.S.F., C.N., C.S., S.D., S.O., A.D.M., M.M.M., I.A. and S.D.; writing—C.S.F.; writing—review and editing, A.D.M. and M.M.M.; project administration, S.O., A.D.M. and I.A.; funding acquisition, S.O., A.D.M. and I.A. All authors have read and agreed to the published version of the manuscript.

**Funding:** This research was funded by POISE/CIG 03 4436 FSE OO1051 under typology 3. 16, Financial and Technical Support to Non-Profit Civil Society Organizations, under Portugal 2020, specifically Priority Axis 3.

**Institutional Review Board Statement:** The study was conducted in accordance with the Declaration of Helsinki and approved by the Ethics Committee, Nursing School of Porto (ADHOC11132022).

**Informed Consent Statement:** Informed consent was obtained from all subjects involved in the study.

**Data Availability Statement:** Not applicable.

**Acknowledgments:** We would like to thank all older persons who took part in the different stages of the board game.

**Conflicts of Interest:** The authors declare no conflict of interest.

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
