# Peer review of "“The Ark of Rights”: Development of a Board Game to Empower Older Adults Regarding Their Rights"

_2673-9259, doi:10.3390/jal3010009_

Round 1
Reviewer 1 Report
The paper describes the procedures used in developing the board/card game “Ark of Rights” and measures its usability. Theoretical background, procedures involved, and results gathered are presented in orderly and understandable fashion. The paper presents an application of usable tool for improving elderly interactions. Unfortunately, the most interesting information, how much of an improvement considering participant’s knowledge of human rights does it provide, is not measured. This was not intended as a research outcome either, which I find is the major omission of study design.
The paper we have is a correct study of target group’s usability. After some major revisions in the introduction and discussion section (as stated below), I could recommend the paper for publishing.
[lines 47-55] this block needs some major rephrasing or usage of direct quoting according to the TurnItIn report (a document is attached)
[59-61] this block needs some major rephrasing or usage of direct quoting according to the TurnItIn report
[62-65] this block needs some major rephrasing or usage of direct quoting according to the TurnItIn report
[84-91] this block needs some major rephrasing or usage of direct quoting according to the TurnItIn report
[112] a group of experts that make the working group – are they the same as authors or are they a different group of people? What sort of experts are they, then? This sentence should be more precise.
[114-125] this block needs some major rephrasing or usage of direct quoting according to the TurnItIn report
[122] “thaemphasize” >> that emphasize
[160-172] please ignore this block on rephrasing; it’s technical jargon and it’s expected to repeat across various papers
[228] put this sentence at the end of line 219; this explains the reason for gaming procedures you developed
[table 1] a) it would be more informative if you used stacked horizontal bars for data representation than using a table format; b)if you decide to stick with the table, either frequencies or percentage should be used, not both; c) complete table should be on the same page, breaking a table header on one page, and the results on the other make it difficult to read
[249-251] remove blank lines, use paragraph properties instead
[252-255] depending how you chose to present the results from table 1, this part should be consistent with it (either frequencies or percentage)
[278-285] this block needs some major rephrasing or usage of direct quoting according to the TurnItIn report
[292-317] this block needs some major rephrasing or usage of direct quoting according to the TurnItIn report

Author Response
Dear Editor and Reviewers,
We are grateful for the feedback and suggestions made during our manuscript review. The changes made have improved the quality of our manuscript, and in the document we provide our point-by-point response to each of the comments.
Best Regards
CSF

Reviewer 2 Report
The article raises a very important issue for the elderly. The authors presented their results in an interesting way. In the INTRODUCTION part, they describe the theoretical foundations of the issues of seniors' rights. The authors also thoroughly describe the research methodology and the selection of the study group. 74 seniors were included in the research, I think that this number is sufficient for this type of research. The research results were presented correctly and clearly. In the discussion, the authors correctly confronted the results of their own research with the results of other authors.
Summing up, I think that the article in this form is suitable for publication in the journal.
I only have one technical note: in references 6, 15, 22 the names of the journals should be capitalized (International Journal of Environmental Research and Public Health; JMIR Serious Games).
Author Response
Dear Editor and Reviewers,
We are grateful for the feedback and suggestions made during our manuscript review. The changes made have improved the quality of our manuscript, and in the document we provide our point-by-point response to each of the comments.

Reviewer 3 Report
-------------------------------
* # Review of the Manuscript Number:
jal-2188651 for the Journal of Ageing and Longevity journal.
* # Title:
"Ark of Rights": Development of a board game to empower older adults regarding their rights
* ## Remarks:
Initially, I'd like to express my gratitude to the authors and editors for providing me with the chance to evaluate this manuscript.
The authors's main objective was to to outline the creation process of a board game named "Ark of Rights"® which was designed to educate older adults about their rights. They evaluated the game using the European Portuguese Validation of the System Usability Scale (SUS).
The subject of the research fits well to the scope of the Journal of Ageing and Longevity journal. The topic is of interest to researchers and practitioners that involved in studying and creating games for older people.
The manuscript is adequately based on the relevant scientific literature and the research objectives are suitably justified. While the contribution is somewhat limited, it is clearly stated. The manuscript is well-structured and the research is presented in a clear and comprehensible manner.
I have, however, some concerns about the scientific soundness of the paper. Among the major problems there are:
[1]
The authors used the SUS questionnaire for assessing the usability of the board game, however the SUS survey was developed for software applications. The authors should comment on this issue.
[2]
The SUS questionnaire is standardized, thus the authors should discuss the results in this context.
[3]
The authors should extend the basic statistics parameters provided - for example adding kurtosis, skewness etc.
[4]
The authors should provide some formal statistical verification of the obtained empirical data apart from the basis descriptive statistics.
[5]
In section 3.3. Pre-test the authors could be more specific and provide some more information on the changes in contents and the language used in the cards at the pre-test phase of the game development.
[6]
Page 3, Line 122:
"... aesthetics thaemphasize them clearly to 122 players, ..." -> rather: "... that emphasize ..."
[7]
Section 3.4. "Ark of Rights" boardgame, Page 5, Line 195:
"6 to 8 players can play the game ..." The authors should consider starting the sentence without the number.
[8]
Section: 3.5. Application and evaluation of the game, page 6
The font size is smaller than in the rest of the paper.
* ## Recommendation
Overall, I recommend this paper to be published in the Journal of Ageing and Longevity journal if the above concerns are satisfactory addressed.
-------------------------------
Author Response

(The authors gave the same response as above.)

Round 2
Reviewer 1 Report
--
Author Response
Dear Editor and Reviewer,
We appreciate the comments.
We added some information to the table and content of the article.
Added a paragraph to the discussion on SUS.
However, we consider that the study is only descriptive, this being the objective of the article, to describe the stages of the development of a game. And not an analytical study, reserved for the next step that is taking place in assessing the impact of the game's effect.
Given the authors' extensive experience in game development, we believe the reviewers will be in agreement on this point.
We took the opportunity to attach the translation certificate.

Reviewer 3 Report
* # Second Review of the Manuscript Number:
jal-2188651 version 2 for the Journal of Ageing and Longevity journal.
* # Title:
"The Ark of Rights": Development of a board game to empower older adults regarding their rights
* ## Remarks:
Firstly, I would like to convey my appreciation to the authors and editors for granting me again the opportunity to review this manuscript.
The authors have significantly improved the paper, however they have not addressed all the issues I have mentioned in my previous review. Moreover, they did not use my numberring, which made it more difficult to follow the changes they made. It would also be easier to verify the authors' responses, if they provide detailed information on how have they addresses the raised issues in the response file. The following remarks have not been addressed:
[2]
The SUS questionnaire is standardized, thus the authors should discuss the results in this context.
[3]
The authors should extend the basic statistics parameters provided - for example adding kurtosis, skewness etc. Right now, the changes are rather minor without much discussion of the obtained reults.
[4]
The authors should provide some formal statistical verification of the obtained empirical data apart from the basis descriptive statistics.
* ## Recommendation
Overall, I recommend this paper to be published in the Journal of Ageing and Longevity journal if the above concerns are satisfactory addressed.
-------------------------------
Author Response

(The authors gave the same response as above.)

Round 3
Reviewer 3 Report
-------------------------------
* # Third Review of the Manuscript Number:
jal-2188651 version 3 for the Journal of Ageing and Longevity journal.
* # Title:
"The Ark of Rights": Development of a board game to empower older adults regarding their rights
* ## Remarks:
The authors have made only minor changes and they have not addressed the issues I have mentioned in my previous reviews. Moreover, they did not use my numberring, which made it more difficult to follow the changes they made. It would also be easier to verify the authors' responses, if they provide detailed information on how have they addresses the raised issues in the response file. The argument that the study is only descriptive does not persuade me as the authors used the formal questionaire, provided their results, analyzed them and drawn conclusions. Without the questionnaire results properly presented and formally analyzed I think that the paper should not be published as its contribution is insufficient. Still, the following remarks have not been properly addressed:
[2]
The SUS questionnaire is standardized, thus the authors should discuss the results in this context.
[3]
The authors should extend the basic statistics parameters provided - for example adding kurtosis, skewness etc. Right now, the changes are rather minor without much discussion of the obtained reults.
[4]
The authors should provide some formal statistical verification of the obtained empirical data apart from the basis descriptive statistics.
* ## Recommendation
Overall, I recommend this paper to be published in the Journal of Ageing and Longevity journal if the above concerns are satisfactory addressed.
Author Response
Dear Editor and Reviewer,
We appreciate the comments, but we have already answered these questions.
We added some information to the table and content of the article.
Added a paragraph to the discussion on SUS.
However, we consider that the study is only descriptive, this being the objective of the article, to describe the stages of the development of a game. And not an analytical study, reserved for the next step that is taking place in assessing the impact of the game's effect.
Given the authors' extensive experience in game development, we believe the reviewers will be in agreement on this point.
We took the opportunity to attach the translation certificate.